# Epidemiological Study of Adenoid Cystic Carcinoma and Its Outcomes: Insights from the Surveillance, Epidemiology, and End Results (SEER) Database

**DOI:** 10.3390/cancers16193383

**Published:** 2024-10-03

**Authors:** Mohamed Rahouma, Sherif Khairallah, Massimo Baudo, Shaikha Al-Thani, Anas Dabsha, David Shenouda, Abdelrahman Mohamed, Arnaldo Dimagli, Magdy El Sherbiny, Mona Kamal, Jonathan Villena-Vargas, Oliver S. Chow

**Affiliations:** 1Cardiothoracic Surgery Departments, Weill Cornell Medicine, Box 110, New York, NY 10065, USA; smk4005@med.cornell.edu (S.K.); massimo.baudo@icloud.com (M.B.); anas.dabsha@hotmail.com (A.D.); ard2014@med.cornell.edu (A.D.); jov9069@med.cornell.edu (J.V.-V.); osc4001@med.cornell.edu (O.S.C.); 2Surgical Oncology Department, National Cancer Institute, Cairo University, Cairo 12613, Egypt; rahmannci@yahoo.com (A.M.); sherbinymi@yahoo.com (M.E.S.); 3Cardiac Surgery Department, Spedali Civili di Brescia, University of Brescia, 25121 Brescia, Italy; 4Biology Department, New York Institute of Technology, New York, NY 11568, USA; shenoudadavid17@gmail.com; 5Symptom Research Departments, The University of Texas MD Anderson Cancer Center, Houston, TX 77030, USA; mkjomaa@mdandrson.org

**Keywords:** adenoid cystic carcinoma, SEER database analysis, geographical variation, surgery, oncology, survival

## Abstract

**Simple Summary:**

This epidemiological study of adenoid cystic carcinoma (ACC) using data from the SEER system aims to investigate independent predictors of late mortality, including factors such as age, stage, and tumor location, as well as racial differences, geographical distribution, and lack of social support (being unmarried). A total of 5150 patients were identified. Our study revealed that stage, tumor location in the thoracic region, and treatment modalities, in addition to geographical distribution (Western region) and lack of social support (being unmarried), were identified as independent predictors of late mortality. While the SEER data are not designed to explain why disease patterns occur, they provide valuable insights into health-related issues that cannot be overlooked. Further studies are needed to determine why the Western region of the USA is associated with poorer survival compared to the Northeast.

**Abstract:**

Objective: Adenoid cystic carcinoma (ACC) is a rare malignant tumor that mainly arises in the head and neck area. We aimed to compare the long-term survival of patients with ACC based on their geographic regions within the United States using the Surveillance, Epidemiology, and End Results (SEER) registry data. Methods: We queried the SEER database to evaluate the geographic distribution of ACC patients based on inpatient admissions. The states included in the study were divided into four geographical regions (Midwest, Northeast, South, and West) based on the U.S. Census Bureau-designated regions and divisions. Demographic and clinical variables were compared between the groups. Kaplan–Meier curves and Cox regression were used to assess late mortality. Results: A total of 5150 patients were included (4.2% from the Midwest, 17.2% from the Northeast, 22.5% from the South, and 56.1% from the West regions). The median follow-up was 12.3 (95% CI: 11.6–13.1 years). Median overall survival was 11.0 (95% CI: 9.2-NR years), 14.3 (95% CI: 12.4–16.4 years), 11.3 (95% CI: 9.7–14.8 years), and 12.0 (95% CI: 11.3–13.0 years) for Midwest, Northeast, South, and West regions, respectively. In multivariable analysis, older age, male sex, thoracic cancer, the presence of regional and distal disease, receiving chemotherapy, not undergoing surgical resection, and being treated in the West vs. Northeast region were found to be independent predictors of poor survival. We identified a significant survival difference between the different regions, with the West exhibiting the worst survival compared to the Northeast region. Conclusions: In addition to the well-known predictors of late mortality in ACC (tumor location, stage, and treatment modalities), our study identified a lack of social support (being unmarried) and geographic location (West region) as independent predictors of late mortality in multivariable analysis. Further research is needed to explore the causal relationships.

## 1. Introduction

Adenoid cystic carcinoma (ACC) was first described by Robin, Lorain, and Laboulbene in 1853, yet to this day, it remains an enigmatic tumor with unpredictable behavior and a dismal prognosis [1,2]. ACC is a rare malignant tumor with a reported incidence of 3–4.5 per million [1]. It arises mainly from minor and major salivary glands, representing 10–25% of malignant tumors of salivary glands [3,4]. The most common site for ACC is the head and neck area, yet it only constitutes 1% of all head and neck malignancies. Other reported areas include the tracheobronchial tree, breast, skin, lacrimal gland, female genital tract, and prostate [4,5].

The course of ACC is indolent but locally aggressive and has a propensity for local recurrence [6]. ACC is notorious for perineural spread, which may explain the likelihood of local and distant spreads and the tendency to recur [7]. The treatment of ACC involves a complete surgical resection with wide surgical margins and, in most cases, adjuvant radiotherapy. Several studies have addressed the role of chemotherapy and targeted therapy in treating ACC. However, there remains to be a consensus on their validity in the primary setting, as they are reserved for advanced and palliative cases [1,6]. Due to ACC’s slow-growing nature and multiple recurrences along its course, data on long-term survival and prognostic factors have been inconsistent [3,7,8].

Studies have illustrated that analyzing the geographic variation among cancer patients can aid in understanding and analyzing cancer risk factors and prognosis [9]. The geographic distribution of patients reflects various relevant factors, such as socioeconomic status, income, health insurance, the availability of resources, the proximity of healthcare services, exposure to carcinogens, and population characteristics (such as median age, gender, education, and race) [10]. Therefore, our aim is to conduct an epidemiological study of ACC to examine the various factors affecting survival, including geographic distribution and race, as well as to analyze the trends in ACC incidence over the study years from the SEER database.

## 2. Materials and Methods

### 2.1. Data Sources

The SEER database of the National Cancer Institute (NCI) was reviewed to identify geographic variations among ACC patients based on inpatient status. The public-use version (https://www.cdc.gov/cancer/uscs/technical_notes/contributors/seer.htm (accessed on 5th December 2022)) of the SEER Research Plus data was used, which is an extended form of the SEER data and contains survival and treatment details. This version was released in November 2021 and includes patients from 2000 to 2019.

### 2.2. Study Population and Inclusion Criteria

Patients with adenoid cystic carcinoma as a single primary tumor (sequence number = 0 or 1) were included, as survival in patients with multiple primary tumors could not be ascribed to a single anatomical cancer site. Additionally, patients were excluded if the follow-up data were missing. Furthermore, the US states were classified into four geographical regions using the U.S. Census Bureau-designated regions and divisions for the geographic classification, which is broadly used for data collection and analysis (Appendix A) [11]. The geographic region was identified based on the inpatient admission site and not on residency.

### 2.3. Study Variables

The following baseline demographics and clinical variables were included in our analysis: age, sex, race, ethnicity, year of diagnosis, tumor site, stage, and treatment modalities (surgery, chemotherapy, or radiotherapy). Relevant socioeconomic details such as marital status, median income, area (nonmetropolitan or metropolitan), SEER registry state, and geographical region were also included in the analysis. Tumor sites were recorded based on the International Classification of Diseases for Oncology (ICD-O-3). We grouped them into head/neck, thoracic, breast, genitourinary, and miscellaneous tumors. Survival in months and vital status (if the patient was alive or dead at the last follow-up) were also retrieved. The outcomes of interest were overall survival (OS) and cancer-specific survival (CSS). The primary variables of interest/exposures in this epidemiological study were tumor site (e.g., thoracic vs. head/neck) and geographic region, while covariates included factors such as age, sex, race, and treatment modalities.

### 2.4. Statistical Analysis

All analyses were performed using R version 4.4.1 within RStudio. Baseline characteristics were compared between the four included geographical groups. Categorical variables such as sex, race, tumor site, and geographic area were presented using frequencies and percentages, and the Chi-squared test or Fisher test was used to compare them accordingly. Continuous variables such as age and year of diagnosis were tested for normality using the Anderson–Darling normality test. If they were normally distributed, they were presented as the mean and standard deviation and compared using an analysis of variance (ANOVA). If they were not normally distributed, they were presented as the median and interquartile range (IQR) and compared using the Kruskal–Wallis test. The overall survival was estimated from the date of diagnosis to the date of death from any cause or to the last follow-up. Cancer-specific survival was estimated from the date of diagnosis until the date of death from cancer or until the last follow-up. Three- and six-year survival rates were estimated using a Kaplan–Meier curve, and the log-rank test was used to compare survival across the included groups. Univariable and multivariable analysis (MVA) predictors of late mortality, defined as mortality at the end of the follow-up period, were estimated using Cox regression and reported as hazard ratios (HRs) and 95% confidence intervals (95% CIs). Univariable predictors with *p* < 0.05 were included in the MVA. Interaction terms between geographic area, age, and stage were also included. We assessed collinearity among the variables included in the regression model using the Variance Inflation Factor (VIF). All VIF values were within acceptable limits, indicating no evidence of collinearity.

The Northeast served as the reference region, as it was associated with the fewest mortality events. Standardized mean difference (SMD) values, a measure of the effect size, were included to foster a meaningful and accurate interpretation of the results for practical/clinical significance, with a larger SMD indicating a greater effect size. Generally, an SMD of 0.2 is considered a small effect size, 0.5 is a moderate effect size, and 0.8 or higher is a large effect size. All *p*-values < 0.05 were considered statistically significant. All tests for *p*-values were two-sided.

## 3. Results

### 3.1. Demographics

A total of 6111 ACC cases were identified in the SEER database; 959 cases were excluded due to the presence of multiple malignant neoplasms over the patient’s lifetime, and two were excluded due to missing survival data. Therefore, a total of 5150 patients were included in the study analysis. The median age of this cohort was 58 years (IQR: 47–70 years). There were 217 (4.2%) patients from the Midwest, 887 (17.2%) from the Northeast, 1158 (22.5%) from the South, and 2888 (56.1%) from the West regions. Overall, there were 1888 male patients (36.7%). Most of the study population was White (75.9%), followed by Asian (11.2%), Black (10.8%), and others (2.1%). Additionally, the majority had localized disease (77.5%) compared to regional (14.3%) and distant (8.2%) disease (Table 1).

The four regions demonstrated significant differences in median income, residence in metropolitan areas, and marital status. About 4.1% of patients from the West had a median income of <USD 50 K, vs. 13.4%, 0%, and 39.9% in the Midwest, Northeast, and South, respectively (*p* < 0.001). Around 94.6% of West and 99% of Northeast patients were from metropolitan areas vs. 52.5% of Midwest and 74.4% of South patients (*p* < 0.001). In the West, 57.7% were married, vs. 54.7% of patients from the Northeast (*p* = 0.001) (Table 1). The racial distribution among the geographic areas is shown in Appendix A.

### 3.2. Tumor Origin and Treatment

Among all patients, the head and neck area was the most common site (70.1%), followed by the breast (14.1%), while thoracic, genitourinary, and miscellaneous accounted for 6.8%, 2.2%, and 6.8%, respectively (Table 1 and Appendix A). The annual trend of different ACC tumor sites is shown in Figure 1, which shows that head and neck tumors had the highest annual incidence, followed by breast, while genitourinary tumors were the least.

A total of 4355 (84.6%) patients underwent surgery, 621 (12.1%) received chemotherapy, and 2845 (55.2%) received radiotherapy. Additionally, 82.1% of patients from the South underwent surgery vs. 84.3%, 85.8%, and 85.2% of Midwest, Northeast, and West patients, respectively (*p* = 0.067, Table 1). The reasons for the lack of cancer-directed surgery are shown in Appendix A. The treatment modalities among different anatomical sites are shown in Appendix A. The differences between ethnicities among geographical regions are shown in Appendix A.

### 3.3. Survival Analysis

The overall median follow-up was 110 months [95% CI: 107–114]. The median follow-up was 95 months in the Midwest versus 109, 112, and 112 months in the Northeast, South, and West regions, respectively.

Median OS was 132 months (95% CI: 110–not reached (NR)), 171 (95% CI: 149–197), 136 (95% CI: 116–178), and 144 (95% CI: 135–156) for Midwest, Northeast, South, and West regions, respectively.

Six-year OS was 71.8% ± 3.5%, 74.4% ± 1.6%, 67.3% ± 1.5%, and 71% ± 0.9% for Midwest, Northeast, South, and West regions, respectively (*p* = 0.064, Figure 2).

Six-year CSS was 71.8% ± 3.4%, 74.4% ± 1.6%, 67.3% ± 1.6%, and 71% ± 0.9% for Midwest, Northeast, South, and West regions, respectively (*p* = 0.064, Appendix A).

Six-year OS was 86.5% ± 1.4%, 75.3% ± 4.5%, 68.2% ± 0.9%, 60.5% ± 3.0%, and 73.8% ± 2.7% for breast, genitourinary, head and neck, thorax, and miscellaneous (skin, and others) cancer locations, respectively (*p* < 0.0001, Figure 3).

Six-year OS was 73.9% ± 2.0% for Hispanic and 70.4% ± 0.7% for non-Hispanic patients (*p* = 0.14, Appendix A).

In univariable analysis, predictors of late mortality included older age (HR = 1.037 [1.033;1.040], *p* < 0.001), male sex (HR = 1.36 [1.24;1.49], *p* < 0.001), thoracic anatomic location, presence of regional LN involvement (HR = 1.40 [1.25;1.58], *p* < 0.001), presence of distant metastasis (HR = 3.28 [2.89;3.72], *p* < 0.001), treatment with chemotherapy (HR = 2.04 [1.81;2.30], *p* < 0.001), non-married patients (HR = 1.32 [1.20;1.44], *p* < 0.001), median income less than USD 50,000 (HR = 1.16 [1.01;1.33], 0.033), and receiving inpatient treatment in the South (HR = 1.21 [1.05;1.41], *p* = 0.009) or West region (HR = 1.16 [1.02;1.32], *p* = 0.023).

In multivariable analysis, older age (HR = 1.05 [1.04;1.06], *p* < 0.001), male sex (HR = 1.25 [1.14;1.38], *p* < 0.001), thoracic anatomic location, regional LN involvement (HR = 1.63 [1.24;2.15], *p* < 0.001), presence of distant metastasis (HR = 2.55 [1.86;3.51], *p* < 0.001), treatment with chemotherapy (HR = 1.76 [1.54;2.01], *p* < 0.001), non-married (HR = 1.27 [1.16;1.40], *p* < 0.001), and West region (HR = 2.93 [1.57;5.49], *p* = 0.001) were found to be independent predictors of late mortality (Table 2). There was no evidence of collinearity among the included variables in the regression model.

A sensitivity analysis among the surgery subset (*n* = 4355) was performed, and the West region remained associated with worse survival (Appendix A).

## 4. Discussion

ACC is a relatively rare malignancy that accounts for only 0.1% of all malignancies, with the head and neck salivary glands being the most affected site [4]. Studies have reported its incidence in various other anatomical regions, such as the breast, lung, cervix, skin, lacrimal glands, and other sites [12]. A few population-based studies of ACC in the literature investigated the tumor’s clinical behaviors or outcomes based on geographical location [13,14], but the overall trends and outcomes of this malignancy are still not well elucidated.

In this study, we found significant survival differences between the four geographical regions within the USA in univariable and multivariable analyses. Specifically, the West region exhibited the worst survival compared to the Northeast, which served as the reference region. To try to explain these differences, we compared clinicopathological and demographic features among the groups. We identified racial differences between the regions, with a solid standardized mean difference (SMD) of 0.648. There was a higher proportion of Asian patients in the West (87.7% of all Asian patients) and of Black patients in the South (50.8% of all Black patients). Nevertheless, in multivariable analysis, we did not find racial differences to be an independent predictor of late mortality.

Ultimately, whether racial differences truly affect cancer survival remains a subject of debate. A comprehensive review by Esnaola et al. reported that most of the medical literature suggests that differences in cancer outcomes between Black and White individuals are primarily attributed to disparities in cancer care rather than racial variations in the stage at presentation, tumor biology, or treatment response [15]. Conversely, a study conducted by Christopher et al. using the SEER data on the incidence and survival trends of head and neck ACC showed a better 5-year relative survival among White (82.4%) and Black patients (85.23%) compared to Asian/Pacific Islander patients (78.4%). However, this difference diminished over time, with Asian/Pacific Islander patients exhibiting better relative survival rates at 10 and 15 years (65.47%) compared to White (60.42%) and Black patients (56.24%) [16]. Additionally, the literature has described a phenomenon described as the Hispanic paradox, where Hispanic individuals exhibit lower mortality rates compared to their White counterparts [17,18,19,20,21]. However, this phenomenon has gradually decreased since 2010 [22]. In 2023, a study illustrated that the mortality advantage of Hispanics had disappeared mainly due to the COVID-19 pandemic, as Hispanic death rates disproportionately increased [23]. In this study, we did not identify a survival advantage for Hispanic over non-Hispanic patients, even in subgroup analyses based on age, gender, and anatomical tumor location.

The disparities in survival might be attributed to factors that cannot be studied using the SEER Research Plus data, such as insurance type, accessibility of transportation, and hospital volume in the region. Some reports suggest that the survival advantage of one geographic area over others is primarily attributed to socioeconomic status, education level, income, decreased access to care (in rural vs. urban areas), and inequalities between healthcare facilities, rather than racial differences [22,23,24,25,26,27,28,29]. While a prior series demonstrated a survival advantage for patients with private insurance vs. Medicare (HR 0.67 (95% CI 0.52–0.87), *p* = 0.002) [30], we were unable to investigate these potential contributors, as our used SEER version did not contain such data variables. In our study, 4.1% of the population in the West (the worst survival) reported an annual income of <USD 50 K versus 0% in the Northeast area (the best survival). However, it is unlikely that this would thoroughly explain the observed survival differences between geographical regions. We are also unable to discern from SEER data whether patients underwent treatment locally or if they traveled more distantly for their treatment, which could be more common in rare malignancies and contribute to geographic variation in outcomes.

### 4.1. Prognosis

The prognosis of ACC is variable and largely dependent on the tumor’s anatomical location, its grade, and the presence of perineurial invasion [13,31,32]. In our study, patients with ACC originating from the breast area had the best survival, as shown in the Kaplan–Meier curves and the multivariable analysis model, which aligns with Sun et al.’s study, which demonstrated better survival for breast ACC (the 10-year CSS and OS were 87.5% and 75.3%, respectively) [33]. Additionally, Bhattacharyya et al. demonstrated 66% 5-year OS for tracheal ACC, which we included as thoracic ACC and corresponds with the poor survival in our study [11].

It is well established that head and neck ACC frequently exhibits perineural spread, leading to upstaging and altering the management approach. Adjuvant radiation is optimal for patients with perineural spread. A SEER data study by Tasoulas et al. demonstrated higher OS among late-stage submandibular ACC patients who received radiotherapy (HR = 0.64, 95% CI: 0.42–0.98) compared to early stages and other anatomic head and neck sites [34]. Furthermore, recent evidence suggests that radiotherapy may also be associated with improved survival in early-stage disease [35]. For breast ACC, Sun et al. reported better survival for those who received adjuvant radiotherapy after lumpectomy than lumpectomy alone [33]. In our study, receiving no surgery and receiving chemotherapy were independent predictors of increased late mortality on multivariable analysis, which aligns with previously published reports [36,37].

### 4.2. Limitations

While our study queried the SEER database, a highly reliable source of incidence, prevalence, and survival assessment, it still has significant limitations. The SEER data do not include postmortem diagnoses, which could underestimate the prevalence of ACC. Also, important potential confounders, such as insurance status, cannot be assessed, as they are not reported in this database. Treatment regimens and their adverse events are also not reported, so their impact on survival cannot be assessed. SEER also has significant amounts of missing data for tumor, nodal, and metastasis (TNM) staging for ACC, which necessitates using a summarized staging system (local, regional, and distant), which could further limit accuracy. Finally, there are no recurrence data in SEER, which would be informative for a tumor with high recurrence and progression rates. Lastly, studying the impact of the geographic region on survival may change over time due to patient migration.

## 5. Conclusions

Our epidemiological study reinforces well-established predictors of late mortality in ACC, such as tumor location, stage, and treatment modalities, but also identified a lack of social support (being unmarried, HR 1.27, *p* < 0.001) and geographical location (West region, HR 2.93, *p* = 0.001) as potential independent predictors of late mortality using multivariable analysis. While databases like SEER are not designed to explain why disease patterns occur, they are instrumental in highlighting specific health-related problems that warrant further investigation. Our study, therefore, may serve as a starting point for additional research aimed at exploring the relationship between these factors and ACC survival.

## Figures and Tables

**Figure 1 cancers-16-03383-f001:**
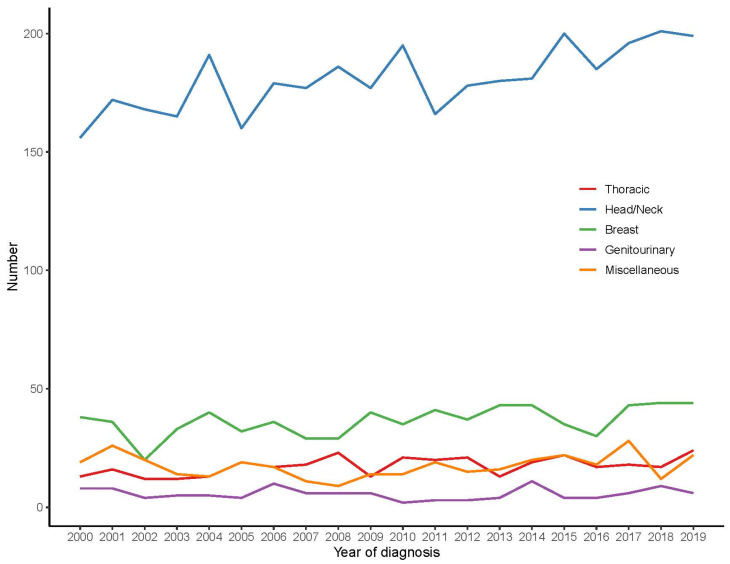
The annual trends of different ACC tumor sites.

**Figure 2 cancers-16-03383-f002:**
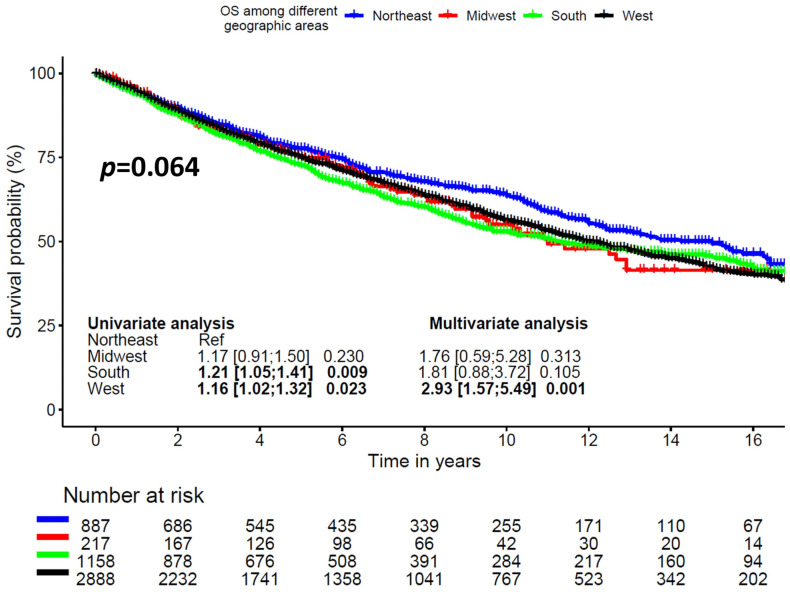
Overall survival according to geographic region.

**Figure 3 cancers-16-03383-f003:**
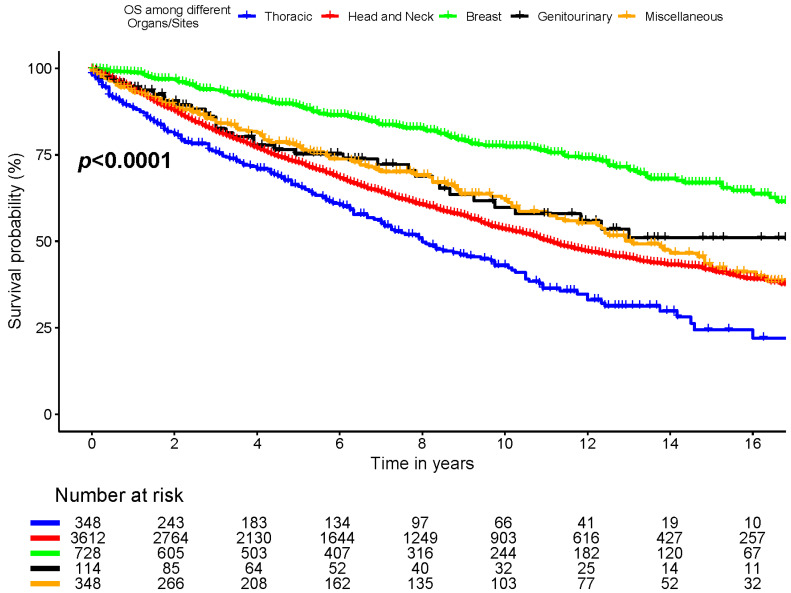
Overall survival among different organs/sites.

**Table 1 cancers-16-03383-t001:** Baseline patient characteristics.

	Overall	Midwest	Northeast	South	West	*p*	SMD
N	5150	217	887	1158	2888		
Age (median [IQR])	58.00 [47.00, 70.00]	58.00 [49.00, 72.00]	58.00 [48.00, 70.00]	59.00 [48.00, 69.00]	58.00 [47.00, 70.00]	0.501	0.056
Male sex (%)	1888 (36.7)	86 (39.6)	310 (34.9)	407 (35.1)	1085 (37.6)	0.25	0.057
Year of diagnosis (median [IQR])	2010.00 [2005.00, 2015.00]	2012.00 [2006.00, 2016.00]	2010.00 [2005.00, 2015.00]	2010.00 [2005.00, 2015.00]	2010.00 [2005.00, 2015.00]	0.107	0.081
Race (%)						<0.001	0.648
Asian	578 (11.2)	0 (0.0)	46 (5.2)	25 (2.2)	507 (17.6)		
Black	556 (10.8)	1 (0.5)	108 (12.2)	283 (24.4)	164 (5.7)		
Other/Unknown	106 (2.1)	0 (0.0)	19 (2.1)	10 (0.9)	77 (2.7)		
White	3910 (75.9)	216 (99.5)	714 (80.5)	840 (72.5)	2140 (74.1)		
Organ/Sites (%)						0.179	0.116
Head/Neck	3612 (70.1)	145 (66.8)	638 (71.9)	781 (67.4)	2048 (70.9)		
Thoracic	348 (6.8)	21 (9.7)	56 (6.3)	89 (7.7)	182 (6.3)		
Breast	728 (14.1)	34 (15.7)	118 (13.3)	187 (16.1)	389 (13.5)		
Genitourinary	114 (2.2)	5 (2.3)	13 (1.5)	25 (2.2)	71 (2.5)		
Miscellaneous	348 (6.8)	12 (5.5)	62 (7.0)	76 (6.6)	198 (6.9)		
Summary stage (%)						0.025	0.1
Distant	422 (8.2)	15 (6.9)	72 (8.1)	90 (7.8)	245 (8.5)		
Regional	738 (14.3)	37 (17.1)	146 (16.5)	132 (11.4)	423 (14.6)		
Unknown/Localized	3990 (77.5)	165 (76.0)	669 (75.4)	936 (80.8)	2220 (76.9)		
Income Less than USD 50 K (%)	609 (11.8)	29 (13.4)	0 (0.0)	462 (39.9)	118 (4.1)	<0.001	0.653
Metropolitan (%)	4587 (89.1)	114 (52.5)	878 (99.0)	862 (74.4)	2733 (94.6)	<0.001	0.742
SEER registry (%)						<0.001	19.287
Alaska Natives	4 (0.1)	0 (0.0)	0 (0.0)	0 (0.0)	4 (0.1)		
California	2218 (43.1)	0 (0.0)	0 (0.0)	0 (0.0)	2218 (76.8)		
Connecticut	263 (5.1)	0 (0.0)	263 (29.7)	0 (0.0)	0 (0.0)		
Georgia	554 (10.8)	0 (0.0)	0 (0.0)	554 (47.8)	0 (0.0)		
Hawaii	103 (2.0)	0 (0.0)	0 (0.0)	0 (0.0)	103 (3.6)		
Iowa	217 (4.2)	217 (100.0)	0 (0.0)	0 (0.0)	0 (0.0)		
Kentucky	301 (5.8)	0 (0.0)	0 (0.0)	301 (26.0)	0 (0.0)		
Louisiana	303 (5.9)	0 (0.0)	0 (0.0)	303 (26.2)	0 (0.0)		
New Jersey	624 (12.1)	0 (0.0)	624 (70.3)	0 (0.0)	0 (0.0)		
New Mexico	107 (2.1)	0 (0.0)	0 (0.0)	0 (0.0)	107 (3.7)		
Seattle (Puget Sound)	313 (6.1)	0 (0.0)	0 (0.0)	0 (0.0)	313 (10.8)		
Utah	143 (2.8)	0 (0.0)	0 (0.0)	0 (0.0)	143 (5.0)		
Married (%)	2908 (56.5)	131 (60.4)	485 (54.7)	626 (54.1)	1666 (57.7)	0.001	0.172
Surgery (%)	4355 (84.6)	183 (84.3)	761 (85.8)	951 (82.1)	2460 (85.2)	0.067	0.054
Radiotherapy (%)	2845 (55.2)	131 (60.4)	506 (57.0)	617 (53.3)	1591 (55.1)	0.151	0.078
Chemotherapy (%)	621 (12.1)	26 (12.0)	108 (12.2)	162 (14.0)	325 (11.3)	0.119	0.042
Survival months (median [IQR])	65.00 [26.00, 123.00]	61.00 [25.00, 104.00]	70.00 [26.50, 128.00]	61.50 [24.00, 118.00]	66.00 [26.00, 124.00]	0.237	0.069
Vital status (Dead (%))	1923 (37.3)	77 (35.5)	298 (33.6)	449 (38.8)	1099 (38.1)	0.064	0.063

SMD = standardized mean difference; IQR = interquartile range.

**Table 2 cancers-16-03383-t002:** Predictors of late overall mortality in the included cohort using Cox regression.

Variable	Univariable Analysis * HR [95% CI], *p*-Value	Multivariable Analysis HR (95% CI), *p*-Value
Age (years)	1.037 [1.033;1.040], <0.001	1.05 [1.04;1.06] <0.001
Male sex (Ref: Female)	1.36 [1.24;1.49], <0.001	1.25 [1.14;1.38] <0.001
Year of diagnosis	1.00 [0.99;1.01], 0.372	---
Race (Black)	Ref.	
● Asian	0.97 [0.80;1.17] 0.720	0.97 [0.79;1.20] 0.799
● Other/Unknown	0.55 [0.34;0.90] 0.018	0.63 [0.38;1.03] 0.067
● White	0.95 [0.83;1.10] 0.530	0.90 [0.77;1.05] 0.177
Organ involved (Ref: Thoracic)	Ref.	
● Head/Neck	0.70 [0.59;0.82] <0.001	0.78 [0.66;0.91] 0.002
● Breast	0.30 [0.24;0.37] <0.001	0.38 [0.30;0.48] <0.001
● Genitourinary	0.53 [0.37;0.76] <0.001	0.45 [0.31;0.65] <0.001
● Miscellaneous	0.60 [0.47;0.75] <0.001	0.62 [0.49;0.78] <0.001
Stage (Ref: Localized)	Ref.	
● Regional	1.40 [1.25;1.58] <0.001	1.63 [1.24;2.15] <0.001
● Distant	3.28 [2.89;3.72] <0.001	2.55 [1.86;3.51] <0.001
Surgery	0.28 [0.25;0.31], <0.001	0.43 [0.37;0.49] <0.001
Radiation	0.70 [0.64;0.76], <0.001	1.02 [0.91;1.14] 0.755
Chemotherapy	2.04 [1.81;2.30], <0.001	1.76 [1.54;2.01] <0.001
Geographic area (Northeast)	Ref.	
● Midwest	1.17 [0.91;1.50] 0.230	1.76 [0.59;5.28] 0.313
● South	1.21 [1.05;1.41] 0.009	1.81 [0.88;3.72] 0.105
● West	1.16 [1.02;1.32] 0.023	2.93 [1.57;5.49] 0.001
Marital status	Ref.	
● Not married	1.32 [1.20;1.44] <0.001	1.27 [1.16;1.40] <0.001
Median income (USD 50,000 or more)	Ref.	
● Less than USD 50,000	1.16 [1.01;1.33] 0.033	0.97 [0.83;1.14] 0.720
Age × Geographic areas [Midwest]	----	0.99 [0.98;1.01] 0.319
Age × Geographic areas [South]	----	1.00 [0.98;1.01] 0.436
Age × Geographic areas [West]	----	0.99 [0.98;1.00] 0.006
Stage [Distant] × Geographic areas [Midwest]	----	1.11 [0.53;2.31] 0.781
Stage [Distant] × Geographic areas [South]	----	0.89 [0.59;1.36] 0.601
Stage [Distant] × Geographic areas [West]	----	0.92 [0.64;1.31] 0.648
Stage [Regional] × Geographic areas [Midwest]	----	1.22 [0.68;2.20] 0.507
Stage [Regional] × Geographic areas [South]	----	0.6 [0.41;0.88] 0.010
Stage [Regional] × Geographic areas [West]	----	0.72 [0.52;0.99] 0.044

* Variables with *p* ≤ 0.05 in univariate analysis were included in multivariate analysis; CI = confidence interval; HR = hazard ratio; NA = not applicable.

## Data Availability

The data that support the findings of this study are available from the corresponding author upon reasonable request.

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
