# Peer review of "Epidemiological Study of Adenoid Cystic Carcinoma and Its Outcomes: Insights from the Surveillance, Epidemiology, and End Results (SEER) Database"

_cancers, 2024, doi:10.3390/cancers16193383_

Round 1

Reviewer 1 Report (Previous Reviewer 2)

Comments and Suggestions for Authors

The comments previously provided were addressed.

No further comments to be adedd.

Author Response

Thank you for your feedback and comments. 

We fully agree with your thoughtful feedback. In response, we have revised the title of our study to: "Epidemiological Study of Adenoid Cystic Carcinoma and Their Outcomes: Insights from the Surveillance, Epidemiology, and End Results (SEER) Database." We have also updated the conclusion to better reflect our study's aim:

"Our epidemiological study not only reinforced well-established predictors of late mortality in ACC, such as tumor location, stage, and treatment modalities, but also identified the lack of social support (being unmarried, HR 1.27, p<0.001) and geographical location (Western region, HR 2.93, p=0.001) as potential predictors of late mortality in the multivariate analysis. While databases like SEER are not designed to explain why disease patterns occur, they are instrumental in highlighting specific health-related issues that warrant further investigation. Our study, therefore, may serve as a starting point for additional research aimed at exploring the causal relationship between these factors and ACC survival." These changes and other changes are highlighted in red in the main manuscript.

We hope these revisions address your concerns

Reviewer 2 Report (Previous Reviewer 4)

Comments and Suggestions for Authors

Thank you for inviting me to review this resubmission of a retrospective cohort study exploring geographical differences in survival among patients with adenoid cystic carcinoma. This study used data from the SEER database, and the authors acknowledge limitations of the database (mostly lacking some variables). While the manuscript is very well written and provides some information about the epidemiology of adenoid carcinomas I do have some comments for the authors, mostly around analysis and interpretation.

The authors state that the objective was to explore differences in survival based on geographical region. Indeed, they did conduct some analyses to address this objective, but they found that several other factors were also related to survival, that arguably are more important for predicting or explaining survival (tumor location) which appears to be the larger focus of this paper. Have the authors considered improving the congruence between the title and objective, and the analyses and interpretation?

The authors have attempted to explore the complexity between geographical region and potential effect measure modifiers and confounders of any association between geographical location and survival, as they present interaction terms (to assess modification) in Table 2. That being said, I am wondering about the models and if confounding was also considered. Similarly, was their collinearity between any variables and geographical region? Collinearity is particularly important given some of the discussion around relationship between race/ethnicity, income, and geographical region. I wonder if the authors have considered more sophisticated analysis to understand the complexity of intersectionality of these variables (such as multilevel analysis of individual heterogeneity and discriminatory accuracy).

Author Response

The authors state that the objective was to explore differences in survival based on geographical region. Indeed, they did conduct some analyses to address this objective, but they found that several other factors were also related to survival, that arguably are more important for predicting or explaining survival (tumor location) which appears to be the larger focus of this paper. Have the authors considered improving the congruence between the title and objective, and the analyses and interpretation?”

Reply: Thank you for your valuable comment. We fully agree with your thoughtful feedback. In response, we have revised the title of our study to: "Epidemiological Study of Adenoid Cystic Carcinoma and Their Outcomes: Insights from the Surveillance, Epidemiology, and End Results (SEER) Database." We have also updated the conclusion to better reflect our study's aim:

"Our epidemiological study not only reinforced well-established predictors of late mortality in ACC, such as tumor location, stage, and treatment modalities, but also identified the lack of social support (being unmarried, HR 1.27, p<0.001) and geographical location (Western region, HR 2.93, p=0.001) as potential predictors of late mortality in the multivariate analysis. While databases like SEER are not designed to explain why disease patterns occur, they are instrumental in highlighting specific health-related issues that warrant further investigation. Our study, therefore, may serve as a starting point for additional research aimed at exploring the causal relationship between these factors and ACC survival." These changes and other changes are highlighted in red in the main manuscript.

We hope these revisions address your concerns.

“The authors have attempted to explore the complexity between geographical region and potential effect measure modifiers and confounders of any association between geographical location and survival, as they present interaction terms (to assess modification) in Table 2. That being said, I am wondering about the models and if confounding was also considered. Similarly, was their collinearity between any variables and geographical region? Collinearity is particularly important given some of the discussion around relationship between race/ethnicity, income, and geographical region. I wonder if the authors have considered more sophisticated analysis to understand the complexity of intersectionality of these variables (such as multilevel analysis of individual heterogeneity and discriminatory accuracy).”

Reply: Thank you for your insightful question. We checked for collinearity among the included variables in our regression model and there was no collinearity.

Thank you for your valuable suggestion regarding the use of more sophisticated analysis methods such as multilevel analysis of individual heterogeneity and discriminatory accuracy (MAIHDA). In our study, we employed Cox proportional hazards regression for the multivariable analysis of late mortality in adenoid cystic carcinoma using the SEER database. This method was chosen because it is well-established for survival analysis and allowed us to account for a range of confounding variables while focusing on the epidemiological aspects of the disease.

We acknowledge that techniques like MAIHDA could provide additional insights, particularly in exploring the complex interactions and intersectionality of individual-level and group-level variables. However, given the scope of our study, which primarily aimed to investigate epidemiological trends and outcomes, Cox regression provided a robust and appropriate framework for our analysis.

Future research might indeed benefit from applying MAIHDA, especially in studies specifically designed to delve into the complexities of intersectionality. We appreciate your suggestion and will consider this approach in future analyses.

Round 2

Reviewer 2 Report (Previous Reviewer 4)

Comments and Suggestions for Authors

Thank you for asking me to review the revised draft of this manuscript. I appreciate the authors making some changes to try to address my concerns with previous drafts.

Unfortunately, many of my prior concerns remain. The slight change in objective was not reflected throughout the manuscript (methods, results, discussion) and there was not demonstration of the presence or absence of collinearity among the variables explored.

There were also some minor comments:

·      In eligibility criteria, were patients with all tumours included? if not please specify the type of tumour in the eligibility criteria.

·      More information about variables – for example what type of variable were they (categorical, continuous, binary)? Delineate outcome vs. exposure vs. covariates.

·      Why were continuous variables automatically reported using non-parametric statistics?

·      There are typos throughout

Comments on the Quality of English Language

The paper is well written, from a grammatical point of view. There are typos throughout.

Author Response

Comments and Suggestions for Authors

Thank you for asking me to review the revised draft of this manuscript. I appreciate the authors making some changes to try to address my concerns with previous drafts.

Unfortunately, many of my prior concerns remain. The slight change in objective was not reflected throughout the manuscript (methods, results, discussion) and there was not demonstration of the presence or absence of collinearity among the variables explored.

Reply: Thanks for your revision. We assessed collinearity among the variables included in the regression model using the Variance Inflation Factor (VIF). All VIF values were within acceptable limits, indicating no evidence of collinearity. We added this to the methods and added the following to the results as well “No evidence of collinearity among the included variables in the regression model.”

There were also some minor comments:

  • In eligibility criteria, were patients with all tumours included? if not please specify the type of tumour in the eligibility criteria

Reply: Thanks for your input. We included any patient with adenoid cystic carcinoma and we excluded patients who had multiple primaries as we mentioned in section 2.2.: ” Patients with adenoid cystic carcinoma as a single primary tumor (sequence number = 0 or 1) were included, as survival in patients with multiple primary tumors could not be ascribed to a single anatomical cancer site.”

  • More information about variables – for example what type of variable were they (categorical, continuous, binary)? Delineate outcome vs. exposure vs. covariates.

Reply: Thanks for your precious and meticulous revision. We edited the methods section and added that. Changes “Categorical variables such as sex, race, tumor site, and geographic area were presented using frequencies and percentages, and the Chi-squared test or Fisher test were used to compare them accordingly. Continuous variables such as age and year of diagnosis were tested for normality using the Anderson-Darling normality test. If they were normally distributed, they were presented as the mean and standard deviation and compared using analysis of variance (ANOVA). If they were not normally distributed, they were presented as the median and interquartile range (IQR) and compared using the Kruskal-Wallis test.”. 

We added the following to the methods “The outcomes of interest were overall survival (OS) and cancer-specific survival (CSS). The primary variables of interest/exposures in this epidemiological study were tumor site (e.g., thoracic vs. head/neck) and geographic region, while covariates included factors such as age, sex, race, and treatment modalities.”

  • Why were continuous variables automatically reported using non-parametric statistics?
  • There are typos throughout

Reply: Thanks for your input. For continuous variables as age we tested for normality using “Anderson-Darling normality test “rather than Shapiro test as number of included cases was > 5000. Consequently, we used non-parametric statistics for them. We edited the methods to be “Continuous variables such as age and year of diagnosis were tested for normality using the Anderson-Darling normality test. If they were normally distributed, they were presented as the mean and standard deviation and compared using analysis of variance (ANOVA). If they were not normally distributed, they were presented as the median and interquartile range (IQR) and compared using the Kruskal-Wallis test.”

Comments on the Quality of English Language

The paper is well written, from a grammatical point of view. There are typos throughout.

Reply: Thank you for your feedback. We have carefully reviewed the entire manuscript and proofread the text to correct any typos.

This manuscript is a resubmission of an earlier submission. The following is a list of the peer review reports and author responses from that submission.

Round 1

Reviewer 1 Report

Comments and Suggestions for Authors

Brief summary

The paper is very interesting, for the number of patients described, as adenoid cystic carcinoma is a quite rare salivary gland malignancy, so that only small series are typically found in the literature.

This is a clear paper, with structured and solid analysis of the parameters studied.

The results are reproducible, and the conclusions are consistent with the thesis and argument presented.

The conclusions are interesting and add advances in the current scientific knowledge.

No ethical problems are found in this study

I would like to make some suggestions and I have few questions

General concept comments

I suggest to specify in the summary and in the discussion that the different regions studied in this paper referred to USA.

In the introduction ACC represents 10% of all salivary gland malignant tumours, in the discussion the percentage is 25% referred to the minor salivary gland only. I suggest to described the percentage of ACC referring to salivary gland in general and then to both major and minor salivary gland.

specific comments:

- line 41: I suggest to modify the sentence in “adenoid cystic carcinoma was first described by Name of the authors in 1853-1854”

- line 45: I suggest to change the second “salivary gland” into “in this anatomical region”

_ line 45*46: I suggest to delete this sentence “It can also arise from lacrimal glands and upper aerodigestive tract glands “ as you described the different anatomic localisation in line 47-48 – you can add here reference n. 4

- line 49: maybe you can change “destructive behaviour” into “local aggressive”

Comments on the Quality of English Language

English level is good 

Reviewer 2 Report

Comments and Suggestions for Authors

Rahouma and colleagues presented a Surveillance, Epidemiology, and End Results (SEER) registry analysis on adenoid cystic carcinoma arising from different primary subsites, including both the most frequent affected head and neck region and non-head and neck region.

Even though the underlying idea is appreciable, the present research paper presents some not negligible weaknesses:

·       It should be specified in materials and methods section which states were included in the 4 regions sub-classifications

·       It should be acknowledged that head and neck and non-head and neck primary subsites represent different tumours, not only in incidence but also in management due to the complexity of surgical strategies in the head and neck district and the consequent different prognosis depending on primary treatment outcome

·       Also, thoracic ACC survival outcome is mainly related to whether the patient can undergo radical upfront surgery  

·       Given the rarity of the disease, it would be of great interest to analyse whether being treated in a referral institution impact on survival outcome. It is well known that centralization in referral centres with high expertise is crucial in the management of rare cancers.

·       Different stage at diagnosis I the paper are categorized as “localized”, “regional” and “distant”. This grouping system is vague and do not reflect the proper stage grouping as for AJCC/TNM 8th edition staging system

·       There are no data regarding disease recurrence (either locoregional or distant). ACC

·       The authors present the analysis of potential predictors for late mortality, but nowhere in the text is clarified what they meant by “late”.

·       In the analysis of potential mortality predictors are included as single variable: “chemotherapy without surgery” and “distant metastasis”. However, being treated with upfront chemotherapy is proxy for advanced metastatic disease at diagnosis.

Reviewer 3 Report

Comments and Suggestions for Authors

A fine analysis of the SEER database, nothing to add compared to the other two reports.

It should be specified in materials and methods section which states were included in the 4 regions sub-classifications

This can be easily accomplished.

·       It should be acknowledged that head and neck and non-head and neck primary subsites represent different tumours, not only in incidence but also in management due to the complexity of surgical strategies in the head and neck district and the consequent different prognosis depending on primary treatment outcome

Same, just a further specification in the discussion

·       Also, thoracic ACC survival outcome is mainly related to whether the patient can undergo radical upfront surgery  

This is obvious for the readers

·       Given the rarity of the disease, it would be of great interest to analyse whether being treated in a referral institution impact on survival outcome. It is well known that centralization in referral centres with high expertise is crucial in the management of rare cancers.

·       Different stage at diagnosis I the paper are categorized as “localized”, “regional” and “distant”. This grouping system is vague and do not reflect the proper stage grouping as for AJCC/TNM 8th edition staging system

True but when you pool cancers from different sites it is not easy to compare the various staging systems; also in the HN area, major versus minor salivary gland tumors have very different TNM 

·       There are no data regarding disease recurrence (either locoregional or distant). ACC

this can be added

·       The authors present the analysis of potential predictors for late mortality, but nowhere in the text is clarified what they meant by “late”.

just a small clarification

·       In the analysis of potential mortality predictors are included as single variable: “chemotherapy without surgery” and “distant metastasis”. However, being treated with upfront chemotherapy is proxy for advanced metastatic disease at diagnosis.

true, a distinct analysis can be performed

Reviewer 4 Report

Comments and Suggestions for Authors

Thank you for inviting me to review this manuscript. The authors explore differences in incidence/prevalence of adenoid cystic carcinoma (ACC) based on geographical location and differences in outcomes using a large national database (SEER). The authors should be commended for a nicely written manuscript that examines an emerging topic of interest. I do have some comments for the authors to consider, listed below.

1.    The authors note in the introduction that little is known about the association between geographical location and ACC, and interestingly justify the importance of geographical location due to it being a surrogate measure of other social determinants of health. I wonder if the authors could rely on studies from other types of cancer (breast, head and neck, etc.) to further justify the need for this study - What is known about geographical distribution currently in other cancers?

2.    Methodological considerations. Given the relatively rare incidence of ACC, the authors made a wise choice in their use of the SEER database. However, there are some questions about cohort inclusion/exclusion criteria.

The authors chose to excluded patients with recurrences or a second primary, but noted that this is an important feature of ACC – what is the impact of this selection bias on the outcomes? Further, in the results we see that this meant excluding 1/6 of all cases which is a significant number that may have implications for the findings. Did the authors consider using recurrence as an outcome?

The primary outcome was survival at end of follow-up - how was time at risk accounted for? In the results it was noted that 3 and 6-year survival was calculated which does take into consideration time at risk, but the description of this analysis was missing from the methods.

3.    Interpretation of results. Upon reviewing the results, I am struck by the interesting differences in race and income between geographical regions (Table 1). The authors note that there were no differences between states (line 196), but indeed there was when the reader examines Table 1. How did the authors tease apart the, likely, complicated relationship between geographical region and these variables? Did the authors consider using modelling to understand the role that race and income play on the association between geographical region and survival (I suspect they are effect measure modifiers or confounders based on previous research)?

4.    The findings into context. The authors compare their findings to some findings in the broader field, but a more critical synthesis of the literature and describing the significance of this work would strengthen the discussion. What does this study add to the field? Why was it a novel, critical piece of work? Will it inform future studies?

The authors describe how this study confirms several previous findings, but it would be interesting to place this work in the broader field of cancer and to expand on some of the interesting findings. For example, the findings around race and income. The authors note that a few studies found no differences in survival based on race, but there are other studies that have found the opposite in other types of cancer (doi: 10.1016/j.soc.2012.03.012, doi:10.1001/jamanetworkopen.2023.27429, https://doi.org/10.1200/OP.20.00381).

Minor comments:

Line 133 – I would avoid using the phrase “only …” as this finding was not statistically significant or clinically significant, especially given the number of patients included in this analysis.

Define “late” mortality.

Round 2

Reviewer 2 Report

Comments and Suggestions for Authors

Even though the authors tried to address all the comments provided, most of raised issues were added as "limitation of the study". 

I would not recommend the publication of the present paper since it does not provide any added value to the literature in the field of ACC.

Author Response

Reviewer 2

Even though the authors tried to address all the comments provided, most of raised issues were added as "limitation of the study". 

I would not recommend the publication of the present paper since it does not provide any added value to the literature in the field of ACC.

Reply to the reviewers and editor comments.

Thank you for your comments. With full respect to the reviewers and editors, we believe our study adds valuable insights to the literature on adenoid cystic carcinoma (ACC).

While the SEER database has limitations, which we address in the limitations section, it remains crucial for conducting epidemiological studies like ours that explore disease patterns. Databases like SEER are not intended to provide answers to why disease patterns occur but rather to highlight specific health problems and their patterns that require further investigation. Such studies can serve as a starting point for conducting further studies to assess the possible association between factors and diseases of interest even from more comprehensive databases as National Inpatients Sample (NIS) and National cancer database (NCDB).

In our study, we identified a geographic discrepancy in survival rates, with the Northeast region showing the lowest mortality rate among the four regions examined. Although we attempted to explain this, the limitations of the SEER data prevented a thorough investigation. However, this does not negate the existence of the problem, and further research is needed to explore the reasons behind it.

It is worth mentioning that we addressed predictors of late mortality. Additionally, we have outlined in the main manuscript the criteria used to classify tumors into different regions and why the Northeast was selected as the reference region, given its low mortality rates. To enhance meaningful interpretation and clinical significance, we included Standardized Mean Differences (SMD) in the statistical analysis. We also examined annual trends of different ACC tumor sites and survival differences between racial groups (Hispanic vs. non-Hispanic). We are open upon your precious input to change the title to be “Epidemiological Study of Adenoid Cystic Carcinoma: Insights from The Surveillance, Epidemiology, and End Results (SEER) Database.”

The added paragraphs have been highlighted in red in the main manuscript for your review.

Thank you for considering our response.